# Bioengineering Support in the Assessment and Rehabilitation of Low Back Pain

**DOI:** 10.3390/bioengineering12090900

**Published:** 2025-08-22

**Authors:** Giustino Varrassi, Matteo Luigi Giuseppe Leoni, Ameen Abdulhasan Al-Alwany, Piercarlo Sarzi Puttini, Giacomo Farì

**Affiliations:** 1Fondazione Paolo Procacci, 00193 Roma, Italy; 2College of Medicine, University of Baghdad, Baghdad 10071, Iraq; ameen.a@comed.uobaghdad.edu.iq; 3Department of Medical and Surgical Sciences and Translational Medicine, “La Sapienza” University of Roma, 00135 Roma, Italy; matteolg.leoni@gmail.com; 4Department of Rheumatology, University of Milano, 20122 Milano, Italy; piercarlo.sarziputtini@unimi.it; 5Department of Experimental Medicine, University of Salento, 73100 Lecce, Italy; giacomo.fari@unisalento.it

**Keywords:** low back pain, bioengineering, wearable sensors, electromyography, virtual reality, artificial intelligence, rehabilitation

## Abstract

Low back pain (LBP) remains one of the most prevalent and disabling musculoskeletal conditions globally, with profound social, economic, and healthcare implications. The rising incidence and chronic nature of LBP highlight the need for more objective, personalized, and effective approaches to assessment and rehabilitation. In this context, bioengineering has emerged as a transformative field, offering novel tools and methodologies that enhance the understanding and management of LBP. This narrative review examines current bioengineering applications in both diagnostic and therapeutic domains. For assessment, technologies such as wearable inertial sensors, three-dimensional motion capture systems, surface electromyography, and biomechanical modeling provide real-time, quantitative insights into posture, movement patterns, and muscle activity. On the therapeutic front, innovations including robotic exoskeletons, neuromuscular electrical stimulation, virtual reality-based rehabilitation, and tele-rehabilitation platforms are increasingly being integrated into multimodal treatment protocols. These technologies support precision medicine by tailoring interventions to each patient’s biomechanical and functional profile. Furthermore, the incorporation of artificial intelligence into clinical workflows enables automated data analysis, predictive modeling, and decision support systems, while future directions such as digital twin technology hold promise for personalized simulation and outcome forecasting. While these advancements are promising, further validation in large-scale, real-world settings is required to ensure safety, efficacy, and equitable accessibility. Ultimately, bioengineering provides a multidimensional, data-driven framework that has the potential to significantly improve the assessment, rehabilitation, and overall management of LBP.

## 1. Introduction

LBP affects up to 80% of individuals during their lifetime and remains a major contributor to the global burden of disease [1,2]. As highlighted by the Global Burden of Disease Study, LBP consistently ranks among the leading causes of years lived with disability (YLDs), significantly impacting workforce participation, healthcare expenditure, and overall quality of life [3]. The etiology of LBP is complex and heterogeneous, involving mechanical, neuropathic, inflammatory, and psychosocial components that vary across patients and disease trajectories [4]. This multifaceted nature, coupled with the inherently subjective experience of pain, poses considerable challenges in establishing accurate diagnoses and formulating effective rehabilitation strategies.

Conventional clinical assessments of LBP typically rely on a combination of patient-reported outcome measures (PROMs), physical examination findings, and static imaging modalities. While these approaches provide valuable information, they often lack the objectivity and granularity necessary to evaluate subtle biomechanical impairments or detect functional changes over time [5]. Instruments such as the Visual Analog Scale (VAS) and the Oswestry Disability Index (ODI) quantify perceived pain intensity and disability but offer limited insight into neuromuscular coordination, postural control, or compensatory movement strategies. Additionally, structural imaging techniques, including magnetic resonance imaging (MRI) and computed tomography (CT), frequently demonstrate poor correlation with clinical symptomatology, potentially leading to underdiagnosis or overtreatment in certain patient subgroups [6].

In recent years, the field of bioengineering has emerged as a pivotal discipline bridging these diagnostic and therapeutic gaps. By integrating physiological measurements, biomechanical principles, and computational modeling, bioengineering offers a suite of technologies capable of delivering objective, reproducible, and high-resolution data relevant to spinal function and dysfunction [7]. Among these innovations, wearable sensor systems, particularly those employing inertial measurement units (IMUs), have proven instrumental in capturing continuous data on spinal motion, gait dynamics, and postural variability. IMUs, comprising accelerometers, gyroscopes, and magnetometers, can be affixed to the trunk, pelvis, or lower limbs, enabling three-dimensional tracking of movement patterns and detection of asymmetries or maladaptive motor behaviors that may perpetuate pain [8]. This transition from episodic clinical assessment to continuous, real-world monitoring enhances both diagnostic fidelity and patient engagement.

Surface electromyography (sEMG) represents another cornerstone of bioengineered assessment, facilitating real-time analysis of muscle activation, fatigue, and recruitment deficits often observed in chronic LBP populations [9]. Advances in miniaturization and wireless technology have yielded high-fidelity EMG systems suitable for both clinical and home use, supporting tailored neuromuscular re-education strategies based on dynamic feedback [10].

Motion capture systems and force platforms further contribute to the comprehensive assessment of posture, balance, and functional tasks such as lifting, walking, or trunk flexion-extension. When integrated with biomechanical models and finite element analysis (FEA), these data streams enable the simulation of internal tissue loads and the evaluation of mechanical stress distribution within spinal components [11,12]. Such simulations not only inform diagnostic classification but also predict tissue response to therapeutic interventions, thereby enhancing the personalization of rehabilitation protocols.

In addition, robotic rehabilitation systems have gained traction as tools for providing programmable, adaptive mechanical assistance or resistance during therapeutic movement. These devices promote safe and repetitive motor practice, facilitating motor relearning in patients with impaired mobility, postural instability, or kinesiophobia [13]. Their use is particularly relevant in subpopulations requiring progressive loading or compensatory training within controlled environments.

Advanced imaging methodologies, including MRI-based radiomics and diffusion tensor imaging (DTI), further enrich the bioengineering toolkit. These technologies allow for the extraction of quantitative biomarkers reflective of intervertebral disc composition, facet joint integrity, and nerve tract alterations [14]. When combined with AI and machine learning algorithms, these data can support predictive modeling and phenotypic clustering, leading to earlier identification of high-risk patients and optimized treatment pathways [15].

Thus, bioengineering introduces a multidimensional and technologically sophisticated framework for the assessment and rehabilitation of LBP. By merging real-time monitoring, functional analysis, and computational prediction, it offers an opportunity to move beyond conventional paradigms toward precision-guided, mechanism-based musculoskeletal care. This narrative review was designed to synthesize current evidence and technological advances related to bioengineering applications in the assessment and rehabilitation of LBP.

## 2. Materials and Methods

The methodological approach adhered to accepted principles for narrative synthesis and was informed by the Scale for the Assessment of Narrative Review Articles (SANRA) to ensure clarity, scientific credibility, and completeness [16]. A structured literature search was conducted across multiple databases, including PubMed, Scopus, Web of Science, and IEEE Xplore, to capture a comprehensive array of peer-reviewed publications. The search strategy combined Medical Subject Headings (MeSH) and free-text terms related to LBP (e.g., “low back pain,” “lumbar pain,” “chronic back pain”), bioengineering tools (e.g., “biomechanics,” “wearable sensors,” “surface electromyography,” “motion analysis,” “robotic rehabilitation,” “finite element analysis”), and advanced technologies (e.g., “artificial intelligence,” “digital health,” “radiomics”). Boolean operators (AND, OR) were applied to optimize sensitivity and specificity of the search.

The initial search covered studies published from January 2014 through June 2025, ensuring a focus on recent advances within the last decade. Only articles published in English were considered. Both primary research articles (e.g., clinical trials, cohort studies, feasibility studies, engineering applications) and secondary literature (e.g., systematic reviews, narrative reviews, technical reports) were included, provided they addressed the integration of bioengineering techniques into the diagnostic or rehabilitative management of LBP.

The selection process proceeded in two phases. First, titles and abstracts were screened for relevance. Studies that did not involve LBP, bioengineering applications, or clinical relevance were excluded. In the second phase, full texts were retrieved and reviewed to confirm eligibility. Articles were included if they reported on the development, clinical implementation, or evaluation of technologies related to motion analysis, neuromuscular assessment, biomechanical modeling, wearable monitoring, robotics, imaging analytics, or digital rehabilitation tools in LBP populations. Following the initial literature search, the reference lists of included articles were manually screened to identify additional pertinent studies. This citation tracking approach revealed emerging rehabilitation technologies frequently integrated with bioengineering tools in LBP management, including virtual reality, augmented reality, neuromuscular electrical stimulation, functional electrical stimulation, and tele-rehabilitation platforms. Given their increasing convergence with sensor-based monitoring and AI-driven personalization in contemporary LBP rehabilitation, a supplementary targeted search was conducted using these specific terms in combination with “low back pain” to ensure comprehensive coverage of the bioengineering landscape in LBP management.

Data extraction was performed manually and included information on study type, technological application, patient population (when applicable), clinical or biomechanical outcomes, and key findings. Where appropriate, findings were grouped thematically based on the phase of care (assessment or rehabilitation) and the primary engineering modality used (e.g., sensor-based tracking, imaging, machine learning). Particular attention was given to evidence supporting objectivity, reliability, clinical utility, and patient-centered outcomes.

The synthesis of included literature was qualitative, aiming to capture technological trends, methodological rigor, and translational potential. The review did not involve meta-analysis or quantitative pooling of data, given the heterogeneity in study designs, technologies, and outcome measures. To ensure scientific integrity, the review incorporated only peer-reviewed sources and emphasized findings supported by reproducible methodologies and real-world application. References were curated based on their relevance, methodological quality, and contribution to understanding how bioengineering can support personalized, mechanistically informed care in LBP management.

## 3. Results

The structured literature search yielded 267 records across four databases. After the removal of duplicates, 214 unique articles were screened based on titles and abstracts. Of these, 136 were excluded due to irrelevance, leaving 78 full-text articles assessed for eligibility. Following detailed evaluation, 43 peer-reviewed studies were included in the final qualitative synthesis. The selection process is depicted in the PRISMA-style flow diagram (Figure 1).

The results of this review were synthesized according to the primary domain of bioengineering intervention and its clinical application in LBP management. The included studies covered a wide spectrum of technological tools applied in either the assessment or rehabilitation phase or both (Table 1).

Wearable sensors, especially IMUs, were the most frequently reported tools, used in eight studies [17,18,19,20,21,22,23,24]. These devices demonstrated high utility for real-time, ecological tracking of lumbar spine kinematics, postural sway, and gait asymmetries. Their clinical relevance lies in detecting functional impairment during activities of daily living and enabling remote assessment paradigms.

sEMG featured in six studies, primarily investigating neuromuscular imbalances, muscle fatigue patterns, and motor control strategies [25,26,27,28,29,30]. These findings have direct implications for tailoring trunk muscle training and evaluating therapy response.

Motion capture systems, described in five studies, enabled precise quantification of segmental motion, particularly during standardized tasks such as forward bending, lifting, and walking [31,32,33,34,35]. Such kinematic data were often integrated with ground reaction forces or EMG to create a multidimensional movement profile.

Biomechanical modeling, including FEA, was explored in six studies [12,36,37,38,39,40]. These models provided insight into internal spinal loading, stress distribution across discs and vertebrae, and hypothetical effects of various rehabilitative exercises. This modality is particularly valuable for developing patient-specific therapeutic interventions.

Robotic rehabilitation systems were evaluated in five studies [13,41,42,43,44]. These devices facilitated programmable and progressive trunk stabilization, gait training, or lumbar support, particularly among patients with significant motor impairment or fear-avoidance behaviors. Robotic tools were reported to improve adherence, safety, and motor relearning outcomes.

Advanced imaging techniques were utilized in seven studies [45,46,47,48,49,50,51]. Applications included MRI-based radiomics and DTI, which yielded quantitative biomarkers associated with disc degeneration, Modic changes, and neural tract integrity. These findings may enhance diagnostic stratification and therapy personalization.

Finally, AI and machine learning tools were integrated in six studies [52,53,54,55,56,57]. These algorithms processed multimodal inputs (kinematics, EMG, imaging) to classify LBP phenotypes, predict rehabilitation outcomes, or guide clinical decision-making. Emerging uses of digital twins and predictive modeling were also noted as future-ready innovations.

In summary, the literature demonstrates a robust and expanding role for bioengineering technologies in the management of LBP. Each modality contributes unique insights, and their combined application offers a more objective, individualized, and mechanistically informed framework for both assessment and rehabilitation. To make this narrative review clinically useful, the results have been divided into two sections regarding diagnostic and therapeutic aspects, as will be discussed later.

### 3.1. Bioengineering in the Assessment of LBP

#### 3.1.1. Wearable Sensors

IMUs, which typically integrate accelerometers, gyroscopes, and magnetometers, have emerged as valuable tools in the objective assessment of lumbar spine function and movement dynamics [17,18]. These wearable sensors provide continuous, real-time data on posture, range of motion (ROM), and segmental kinematics across multiple planes of movement [19,20]. Their capacity for high-resolution motion tracking has been extensively validated in both clinical and ambulatory environments, making them particularly suited for assessing functional mobility in individuals with LBP [58,59].

Unlike traditional observational or static assessment techniques, IMU-based systems offer an ecological and dynamic method for evaluating motor behavior during activities of daily living, including sporting and work activities [23]. They can detect subtle deviations from normative movement patterns, including aberrant spinal alignment, asymmetrical gait, and compensatory trunk or pelvic motions, factors often associated with chronic LBP (CLBP), functional disability, or maladaptive coping strategies [17,60]. As a consequence, they allow tracking an individual’s lumbar movement-based phenotype, highlighting the correlations between spinal kinematics and patient-reported outcomes [61]. Additionally, these devices facilitate longitudinal monitoring, enabling clinicians to track rehabilitation progress, quantify treatment effects, and detect early signs of deterioration or improvement with minimal patient burden [62].

Recent developments have extended the functionality of IMUs through the deployment of multi-sensor arrays positioned over the lumbar spine, pelvis, and lower extremities [63]. This configuration allows for a three-dimensional reconstruction of trunk and pelvic kinematics, which is particularly relevant for understanding complex intersegmental coordination and load transfer mechanisms during tasks such as lifting, bending, or walking [18,24]. Moreover, the integration of IMU data with machine learning algorithms and biomechanical modeling frameworks has opened new avenues for personalized diagnosis and intervention [64]. In fact, advanced wearable sensors could facilitate the development of TR protocols and in-presence exercise programs completely individualized according to a patient’s LBP mechanisms and characteristics [65].

Despite their promise, IMU-based systems face several practical constraints in clinical implementation. Sensor drift and calibration errors can accumulate over extended monitoring periods, requiring frequent recalibration that may disrupt clinical workflows [66]. The placement variability of sensors between sessions or different operators can lead to inconsistent measurements, affecting longitudinal comparisons [67]. Additionally, skin motion artifacts, particularly in patients with higher BMI or during dynamic activities, can compromise data accuracy. Battery life limitations restrict continuous monitoring duration, while the need for multiple sensors to capture complex spinal movements increases setup time and patient burden. Furthermore, the lack of standardized protocols for sensor placement and data interpretation across different manufacturer systems limits inter-study comparisons and clinical adoption.

#### 3.1.2. Surface Electromyography (sEMG)

sEMG is a non-invasive bioengineering tool that provides detailed information about the electrical activity of superficial muscles, including the paraspinal and abdominal musculature [28]. Unlike traditional needle electromyography, sEMG is a non-invasive and easy-to-use method that is revolutionizing various fields of applied rehabilitation, particularly in sports rehabilitation [68].

Its application in LBP research and rehabilitation has significantly advanced the understanding of neuromuscular dysfunctions commonly observed in this population [30]. Patients with CLBP often exhibit abnormal muscle recruitment patterns, increased fatigability, and bilateral asymmetries in activation, particularly in the lumbar erector spinae and multifidus muscles [69]. These neuromuscular deficits are believed to contribute to impaired spinal stability, altered movement strategies, and heightened risk of recurrence.

A study by Arvanitidis et al. revealed that patients suffering from CLBP increased sEMG-torque coherence in more cranial lumbar erector spinae regions, while the opposite was observed for the controls (*p* = 0.043), suggesting that pain could be related to compensatory strategies and regional adjustments of this muscle’s oscillatory activity [25]. Recent technological developments have led to the miniaturization and refinement of sEMG systems [70]. Modern wireless sEMG devices now offer improved signal-to-noise ratios, multi-channel recording capability, and real-time data visualization, which facilitate their use in both clinical and ambulatory environments [29]. These systems enable the assessment of dynamic muscle function during a variety of functional tasks, such as gait, trunk flexion-extension, and lifting, providing ecologically valid insights into motor control behavior. In this context, a recent systematic review by Moessenet et al. proposed an interesting muscular activity biomarker model to understand the musculoskeletal factors underlying chronic non-specific LBP [27]. This enables rehabilitation professionals to develop more specific therapeutic strategies for these patients, selecting the target muscles of rehabilitation with extreme precision.

Therefore, the integration of sEMG data into biofeedback platforms and neuromuscular training protocols allows for personalized rehabilitation approaches [71]. Real-time feedback on muscle activation enables patients to consciously adjust their recruitment patterns, promoting more efficient and symmetrical muscle engagement [26,56]. Most importantly, this muscle feedback also allows physiotherapists to monitor and modify the execution of therapeutic exercises, making them more effective and enabling quicker functional recovery [72]. Therefore, sEMG represents a robust and evolving modality for quantifying neuromuscular function in LBP, with strong potential to inform precision-guided rehabilitation strategies.

Clinical implementation of sEMG faces significant technical and practical challenges. Crosstalk from adjacent muscles remains a persistent issue, particularly in the lumbar region where multiple muscle layers overlap, potentially confounding the interpretation of individual muscle activity [73]. Day-to-day variability in electrode placement and skin preparation can result in measurement inconsistencies of up to 20–30% [74]. Adipose tissue thickness significantly attenuates EMG signals, making reliable measurements challenging in overweight patients who constitute a substantial portion of the LBP population [75]. Environmental electromagnetic interference in clinical settings can corrupt signals, requiring careful shielding and filtering. Moreover, the interpretation of sEMG data requires specialized expertise not universally available in clinical settings, and normative databases for pathological populations remain limited, hampering diagnostic utility.

#### 3.1.3. Motion Capture and Gait Analysis

Optoelectronic motion capture systems are widely regarded as the gold standard for high-fidelity analysis of spinal and whole-body kinematics [76]. These systems utilize arrays of high-speed infrared cameras and reflective markers to precisely track three-dimensional joint and segmental movements with sub-millimeter accuracy. Their application in the assessment of LBP has enabled detailed characterization of spinal mobility, intersegmental coordination, and postural adaptations during static and dynamic tasks [8]. When integrated with force platforms and sEMG, motion capture systems offer a comprehensive biomechanical framework capable of evaluating ground reaction forces, joint moments, and muscle activation patterns [77]. This multimodal approach provides robust insight into neuromechanical impairments, particularly in CLBP populations, where altered trunk control, reduced lumbar curvature, and desynchronized movement patterns have been consistently reported [35,57].

Such integration is especially valuable in distinguishing between functional subgroups of LBP and in assessing responses to therapeutic interventions. Accordingly, gait analysis can even constitute a tool for quantifying LBP in a more objective way than traditional self-reported scales. Giglio et al. described an interesting clinical case of LBP relief using radiofrequency for facet joint syndrome, but more importantly, the authors provided an in-depth analysis of the motion changes as pain modification indicators [78].

Despite their diagnostic accuracy, traditional optoelectronic systems are often limited to laboratory settings due to high costs, complex setup requirements, and spatial constraints. However, recent advancements in portable and marker-less motion-capture technologies, such as depth sensors and wearable inertial devices, are facilitating the translation of this modality into clinical practice [77,79]. These emerging systems offer reduced setup time, user-friendly interfaces, and satisfactory accuracy for routine assessment, making them more accessible for physiotherapists and rehabilitation specialists [24,80]. As technological refinements continue, motion capture, particularly when integrated with musculoskeletal modeling and AI-based analytics, is expected to play a central role in personalized movement analysis and precision rehabilitation planning for LBP patients.

Traditional optoelectronic systems, although highly accurate, face significant barriers to clinical adoption due to high costs, dedicated laboratory requirements, lengthy setup times, and issues such as marker occlusion or altered movement patterns [81]. Furthermore, their confinement to controlled laboratory settings compromises real-world applicability. While marker-less approaches offer greater accessibility, they still face challenges with accuracy, clothing interference, and lighting variability.

#### 3.1.4. Biomechanical Modeling and Finite Element Analysis (FEA)

Computational modeling, particularly FEA, has become a pivotal tool in the biomechanical evaluation of spinal function and dysfunction [12,40]. By replicating the anatomical and material properties of spinal components, such as vertebrae, intervertebral discs, ligaments, and musculature, these models enable detailed simulation of spinal load distribution, segmental deformation, and disc mechanics under a range of postural and loading conditions [36]. This approach is especially valuable in LBP research, where internal stress patterns and tissue-level interactions cannot be directly observed in vivo.

Patient-specific finite element models, derived from high-resolution imaging modalities (e.g., MRI, CT), offer an individualized representation of spinal biomechanics, facilitating the prediction of mechanical responses to various rehabilitative exercises, postural corrections, or surgical interventions [38,82]. These simulations can account for anatomical variability, degenerative changes, and altered boundary conditions, providing insights into the biomechanical consequences of clinical decisions such as intervertebral disc unloading, lumbar fusion, or trunk stabilization exercises [83].

Beyond research and diagnosis, FEA serves an important role in clinical education and shared decision-making [84]. Visualization of stress concentrations or deformation patterns allows clinicians to explain pathology and therapeutic rationale in a patient-specific context, thereby enhancing understanding and treatment adherence [85]. Furthermore, when combined with motion capture and electromyography data, FEA models can be dynamically driven to simulate real-life movements, offering a unique perspective on how altered kinematics or neuromuscular control affect spinal tissue loading [86].

Overall, computational modeling bridges the gap between static anatomical imaging and dynamic mechanical function, supporting precision-guided diagnosis, risk assessment, and intervention planning in individuals with LBP.

The clinical translation of FEA faces considerable computational and validation challenges. Creating patient-specific models requires high-resolution imaging and can take several hours of processing time, making real-time clinical decision-making impractical. The accuracy of models heavily depends on assumptions about tissue material properties, which vary significantly between individuals and cannot be directly measured non-invasively [87]. Validation of model predictions against in vivo measurements remains limited due to ethical constraints on invasive measurements. The computational expertise required for model development and interpretation is rarely available in clinical settings. Furthermore, current models often oversimplify complex biological processes such as muscle co-contraction patterns and time-dependent tissue responses, potentially limiting their predictive accuracy for individual patient outcomes.

#### 3.1.5. Imaging Biomarkers and Radiomics

Advanced imaging modalities such as MRI, CT, and musculoskeletal ultrasound have long been cornerstones in the diagnostic workup of LBP [47]. These techniques allow for visualization of structural abnormalities including intervertebral disc degeneration, facet joint arthropathy, spinal canal stenosis, and paraspinal muscle atrophy. However, conventional imaging is often limited by its qualitative nature and weak correlation with symptom severity, particularly in cases of non-specific or CLBP [6,88].

In recent years, the emergence of radiomics (the high-throughput extraction and analysis of quantitative features from medical images) has revolutionized spinal imaging by enabling objective, reproducible characterization of tissue morphology, texture, and intensity patterns [51]. When applied to lumbar spine MRI or CT, radiomic signatures have demonstrated potential to identify Modic changes, stratify disc degeneration, and assess vertebral endplate integrity with greater sensitivity than traditional radiological scoring systems [47,89].

These advancements have been further amplified through integration with machine learning algorithms. Supervised learning models can analyze complex radiomic datasets to classify pain phenotypes, predict rehabilitation or surgical outcomes, and differentiate between nociceptive and neuropathic pain mechanisms [55,90]. For instance, DTI and quantitative T2 mapping have been used to detect microstructural alterations in the spinal cord and adjacent neural tissues, thereby enhancing the diagnostic yield in cases of lumbar radiculopathy or spinal cord injury [91].

Muhaimil et al. [92] carried out a case-control study on 200 MRI patients referred for LBP and found that machine learning and deep learning models could effectively predict lumbar pain by analyzing T2 weighted images of the lumbar spine. Similarly, Climent-Peris et al. [93] conducted a prospective-observational study to build a predictive model using lumbar MRI to identify patients who are more likely to improve their non-specific CLBP through rehabilitation programs.

Moreover, the combination of radiomic data with clinical variables and functional metrics, such as gait or electromyographic patterns, holds promise for building predictive models and digital twins that may support personalized treatment planning in LBP populations [94]. As computational tools and imaging resolution continue to evolve, radiomics stands to significantly improve diagnostic precision, prognostication, and therapy selection in musculoskeletal spine care. An overview of the bioengineering modalities applied in the assessment of low back pain (LBP) is presented in Figure 2.

Radiomic analysis faces several methodological and practical problems. The lack of standardization in image acquisition parameters, feature extraction methods, and analysis pipelines leads to poor reproducibility across institutions [95]. Features extracted from different MRI scanners or even different software versions show substantial variability, limiting the generalizability of predictive models. The “curse of dimensionality” in radiomics, where hundreds of features are extracted from limited patient samples, increases the risk of overfitting and false discoveries. Many radiomic signatures lack biological interpretability, making it difficult for clinicians to understand and trust the results. Additionally, the time and computational resources required for radiomic analysis, combined with the need for specialized software and expertise, present barriers to routine clinical implementation.

### 3.2. Bioengineering in Rehabilitation of LBP

#### 3.2.1. Robotic Rehabilitation Systems

Robotic-assisted rehabilitation has emerged as a promising approach for delivering precision-controlled, repeatable, and adaptable training paradigms to individuals suffering from many chronic diseases, including LBP [13,41,96]. These technologies include active orthoses and lumbar exoskeletons that offer dynamic postural support, promote symmetrical movement execution, and reduce excessive spinal loading during tasks involving lifting, walking, or prolonged standing [13,97]. By offloading the lumbar spine, such systems aim to correct maladaptive motor strategies and facilitate the re-establishment of neuromuscular control, particularly in patients with chronic or recurrent LBP [98].

Recent studies have demonstrated that robotic rehabilitation may enhance treatment adherence, improve trunk stability, and reduce pain severity in both occupational and clinical settings [99,100]. Qi et al. [101] proposed an exoskeleton based on a single compression spring with a lower support moment and higher traction performance, obtaining a reduction in L5/S1 disc pressure without additionally increasing muscle activation, thus relieving LBP. A systematic review published in 2021 stated that back-support exoskeletons and soft robotic suits could drastically reduce LBP and musculoskeletal diseases in workers exposed to physical loads and manual handling tasks [102]. This demonstrates that innovative ergonomics solutions could really counteract the biomechanical factors favoring LBP, but there are various technical challenges and a lack of established safety standards limiting these positive perspectives.

Importantly, advanced exoskeletons allow for real-time adjustment of torque, stiffness, and resistance parameters based on the user’s physical capacity and rehabilitation goals, enabling personalized progression and safety [103]. This level of individualization makes robotic interventions well suited for patients with high levels of fear-avoidance or impaired functional capacity.

Although traditionally limited to research environments, the increasing miniaturization and cost-effectiveness of robotic systems support their potential for integration into outpatient rehabilitation and workplace prevention programs. These devices may be especially beneficial in cases requiring repetitive task training, reconditioning, or biomechanical unloading of the lumbar spine [104]. Despite promising outcomes, robotic rehabilitation systems face substantial implementation barriers. The high initial investment (ranging from $50,000 to over $500,000) and ongoing maintenance costs make them financially prohibitive for many rehabilitation centers. Current exoskeletons are often heavy (5–15 kg) and bulky and can cause discomfort during prolonged use, potentially limiting patient acceptance [105]. The rigid movement patterns imposed by some systems may not accommodate individual anatomical variations or pathological movement strategies. Setup and adjustment time (15–30 min per session) reduces therapy efficiency. Safety concerns, including potential skin irritation, pressure sores, and the risk of falls due to system malfunction, require constant supervision.

#### 3.2.2. Virtual Reality and Augmented Feedback

VR-based rehabilitation integrates immersive virtual environments with real-time biofeedback to actively engage patients both cognitively and physically during therapeutic tasks [106]. The scientific literature grants solid evidence regarding VR-based rehabilitation effectiveness in patients suffering from neurological diseases, such as multiple sclerosis and Parkinson’s disease [107,108].

In the same way, VR has been shown to significantly enhance motivation and treatment adherence in individuals with CLBP [109]. For example, Matheve et al. [110] demonstrated that VR distraction induced hypoalgesia in CLBP patients and reduced kinesiophobia. These systems use visual and auditory feedback loops that facilitate motor learning by reinforcing correct movement patterns and providing immediate corrective cues [111]. Furthermore, VR platforms can simulate real-world functional scenarios, such as bending, reaching, or walking in varying contexts; thereby promoting functional improvements and diminishing fear-related avoidance behaviors common in chronic pain populations [112]. A recent meta-analysis reported that VR-based training produced significant reductions in both pain intensity and pain-related fear, although sustained effects on disability at short-term follow-up were not demonstrated [113].

Moreover, a comparative study by Massah et al. [114] showed that exergames, a clinical application of VR, may have potential therapeutic advantages over traditional core stability exercise interventions for CLBP. The implementation of VR in LBP rehabilitation faces several practical challenges. Cybersickness affects 20–60% of users and can cause nausea, dizziness, and headaches, potentially exacerbating pain symptoms and reducing treatment adherence [115]. The technology gap among older patients, who constitute a large portion of chronic LBP sufferers, creates barriers to adoption and effective use [116]. High-quality VR systems remain expensive ($3000–$10,000), while cheaper alternatives may compromise therapeutic effectiveness. Limited physical space requirements for safe VR use may exclude home-based applications. The lack of haptic feedback in most systems reduces the realism of motor learning experiences. Additionally, prolonged VR use can cause eye strain and postural problems, potentially counteracting therapeutic benefits.

In conclusion, the integration of immersive VR into TR projects can improve LBP treatments for both patients and healthcare providers, but there is still the need to make these technological tools economically accessible and easy to use [117].

#### 3.2.3. Neuromuscular Electrical Stimulation (NMES) and Functional Electrical Stimulation (FES)

NMES has emerged as an effective modality for re-educating inhibited or atrophied trunk muscles in individuals with CLBP [118]. Surface NMES applied to the lumbar multifidus or erector spinae enhances muscle activation, strength, and endurance, with studies demonstrating significant improvements in motor unit recruitment and muscle cross-sectional area (CSA) after multi-week treatment sessions [119]. FES extends this capability by enabling active movement via electrically stimulated contractions of targeted muscle groups, which is particularly beneficial during post-operative recovery or severe deconditioning [117].

Wolfe et al. [120] found that multifidus NMES can effectively reduce LBP, improving muscle activation and counteracting its stiffness. Recent advancements have led to closed-loop NMES/FES systems that dynamically adjust stimulation parameters based on real-time feedback from movement sensors or EMG, optimizing motor responses and reducing muscle fatigue [121].

NMES also represents a useful diagnostic tool. Wattananon et al. [122] investigated the usefulness of the combination of NMES and ultrasound for detecting muscular deficit of the multifidus, which is often involved in CLBP; the authors suggested that NMES could also be a great prognostic instrument in the rehabilitation field, allowing the monitoring of the multifidus response to physiotherapy and exercises. Clinical application of NMES/FES faces several physiological and practical constraints. Patient tolerance to electrical stimulation varies significantly, with some unable to achieve therapeutic muscle contraction levels due to discomfort. Muscle fatigue occurs rapidly with electrical stimulation compared to voluntary contractions, limiting session duration and effectiveness [123]. Adipose tissue significantly increases impedance, requiring higher stimulation intensities that may cause skin irritation or burns. Precise electrode placement is critical but challenging to replicate between sessions, affecting treatment consistency. Contraindications include pregnancy, pacemakers, and skin conditions, excluding significant patient populations. The passive nature of muscle activation may not effectively retrain motor control patterns essential for functional improvement.

#### 3.2.4. Computer-Guided Exercise Programs and Tele-Rehabilitation (TR)

Digitally delivered exercise programs, guided by motion sensors or camera-based systems, offer personalized, real-time instruction and correction for individuals with LBP. These platforms facilitate remote monitoring and ensure adherence to exercise protocols, improving the fidelity and consistency of home-based therapy [124,125]. Embedded bioengineering tools (such as many of those mentioned before in this review) enable dynamic adjustment of exercise difficulty, objective progress tracking, and generation of performance metrics [126]. Consequently, these systems bridge the gap between clinic and home, enhancing patient engagement and treatment efficacy.

It is therefore no coincidence that TR has reached its maximum expansion in recent years, following the onset of the COVID-19 pandemic and the restrictions it entailed to reduce the virus spread [127]. Moreover, TR offers the opportunity to treat and monitor a patient with a musculoskeletal disease, providing many rehabilitation solutions based on advanced technological devices [128]. LBP, especially in its chronic form, represents a pivotal field of TR application.

A recent review states that TR helps to improve treatment adherence and pain relief in LBP patients, even if the high heterogeneity in the use of digital methods poses limitations on conclusive outcomes [129]. A meta-analysis by Lara-Palomo et al. [130] showed that digital drugs aimed at self-maintenance and education are as effective on pain and back-specific functional status as other face-to-face or home-based interventions in patients with CLBP, with significant scientific evidence. Moreover, TR can represent a way to reduce rehabilitation costs and times [131]. Obviously, there are still some concerns about the full dissemination of TR in the treatment of LBP, as well as the other main musculoskeletal disorders. Firstly, there is the need to grant all patients, regardless of their economic possibilities and the geographical area in which they live, access to the technological tools necessary for TR [132]. This requires substantial investments by health systems. Most importantly, TR still needs to be made more efficient to safeguard the sacredness of the doctor-patient relationship, which in its traditional face-to-face form allows for the creation of strong empathy and for therapies to be adapted to the needs of each patient [133]. Nevertheless, TR undoubtedly represents a promising horizon for LBP rehabilitation. The main bioengineering technologies currently applied in the rehabilitation of LBP, including innovations in robotic assistance, VR, neuromuscular stimulation, and TR, are summarized in Figure 3.

Digital rehabilitation platforms face multiple implementation challenges. The digital divide excludes patients without reliable internet access or appropriate devices, potentially exacerbating healthcare disparities. The absence of hands-on assessment and manual therapy components may compromise treatment effectiveness for certain conditions. Technology troubleshooting requirements can frustrate less tech-savvy patients and consume clinical support time. Privacy and data security concerns, particularly with video-based systems, may deter patient participation. The lack of immediate therapist feedback may allow incorrect movement patterns to persist or worsen. Insurance reimbursement policies for telerehabilitation remain inconsistent across regions. Furthermore, maintaining patient engagement and adherence without in-person accountability proves challenging, with dropout rates often exceeding 50% [134].

#### 3.2.5. Advanced Human-Robot Interfaces and Adaptive Control Systems

Recent breakthroughs in human-robot interaction technologies have introduced sophisticated control strategies that significantly enhance the therapeutic potential of robotic rehabilitation systems for LBP. Adaptive torque estimation algorithms represent a fundamental advance in rehabilitation robotics, enabling systems to achieve unprecedented accuracy in force delivery while maintaining robust tracking performance during dynamic therapeutic movements [135]. These algorithms continuously adjust assistance levels based on real-time biomechanical feedback, automatically compensating for patient fatigue, pain-related movement alterations, and individual motor learning rates. For LBP patients, this adaptive capability is particularly valuable, as it allows for precise unloading of spinal structures while encouraging active participation, thereby preventing the muscle deconditioning often associated with passive support systems.

The integration of vision transformer (ViT) architectures into wearable rehabilitation systems has revolutionized environmental awareness and adaptive assistance strategies [136]. These deep learning models process visual information to recognize terrain characteristics, obstacles, and movement contexts, enabling exosuits to preemptively adjust lumbar support and movement assistance parameters. This anticipatory adaptation is crucial for LBP patients who often exhibit heightened vulnerability to unexpected perturbations or surface transitions that could trigger pain exacerbations. The ViT-based systems demonstrate superior generalization across diverse environments compared to traditional computer vision approaches, making them particularly suitable for supporting the transition from controlled clinical settings to complex real-world scenarios.

Biosignal-based human-robot interfaces, particularly those utilizing sEMG envelope processing, have emerged as critical enablers of intuitive and responsive robotic control [137]. These interfaces decode muscle activation patterns with minimal latency, allowing collaborative wearable robots to synchronize seamlessly with users’ movement intentions. For LBP rehabilitation, this technology enables real-time detection of compensatory muscle activation patterns, muscle fatigue onset, and aberrant motor control strategies. The envelope-based signal processing approach offers improved robustness against motion artifacts and electrical noise compared to raw sEMG analysis, enhancing reliability in dynamic rehabilitation contexts. Furthermore, these interfaces can distinguish between voluntary muscle activation and involuntary spasms or guarding behaviors common in chronic LBP, allowing the robotic system to respond appropriately to each scenario.

The convergence of these technologies creates a new paradigm for intelligent, adaptive rehabilitation systems. Modern rehabilitation robots can now combine adaptive torque control with terrain-aware assistance and intuitive biosignal interfaces to provide truly personalized therapeutic interventions. Machine learning algorithms integrate multimodal data streams from these various interfaces to build patient-specific models that predict optimal assistance strategies, identify movement patterns associated with pain provocation, and adapt therapeutic parameters in real-time. This technological synthesis is particularly promising for addressing the heterogeneous nature of LBP, as systems can now adapt not only to anatomical and biomechanical variations but also to psychomotor factors such as fear-avoidance behaviors and movement confidence.

Despite these advances, several challenges remain in translating these sophisticated interfaces to clinical practice. The computational demands of real-time ViT processing and adaptive control algorithms require powerful embedded systems that may increase device cost and complexity. Integration of multiple sensor modalities raises concerns about system reliability, as failure of any component could compromise therapeutic safety. Additionally, the need for initial calibration and periodic recalibration of these adaptive systems may limit their practicality in high-volume clinical settings. Nevertheless, ongoing research in edge computing, sensor fusion algorithms, and automated calibration procedures continues to address these limitations, bringing these advanced human-robot interfaces closer to widespread clinical implementation.

### 3.3. Integrating Artificial Intelligence (AI) and Machine Learning (ML)

AI is fundamentally transforming the assessment and management of LBP by introducing sophisticated data-driven personalization and advanced predictive modeling capabilities. Machine learning methodologies, encompassing both supervised learning algorithms such as support vector machines and random forests, as well as unsupervised clustering techniques, have been successfully implemented to classify and stratify LBP phenotypes through comprehensive analysis of integrated datasets incorporating movement kinematics, neuroimaging biomarkers, psychosocial factors, and behavioral metrics [138,139].

Recent studies have demonstrated that ML algorithms can achieve 79% accuracy in identifying CLBP patients using motion complexity analysis from inertial sensors [140]. These computational approaches not only facilitate early identification of distinct patient subgroups with high likelihood of responding to targeted rehabilitation interventions but also reveal previously hidden biomechanical-clinical associations that traditional analytical methods might overlook [141].

Advanced AI models incorporating disc height measurements and ligamentum flavum hypertrophy analysis have shown significant potential in predicting LBP presence from lumbar MRI or CT imaging data [142]. Recent advancements in digital twin technology represent a paradigm shift, enabling the development of sophisticated virtual replicas of individual patients’ musculoskeletal systems that can simulate therapeutic responses and optimize personalized rehabilitation protocols in real time [143,144].

Current applications of digital twin systems in musculoskeletal medicine include optimizing exercise and rehabilitation monitoring, analyzing joint mechanics for personalized surgical techniques, and predicting post-operative outcomes [145]. Deep learning architectures, particularly convolutional neural networks, have shown remarkable success in automated analysis of medical imaging data, such as spine MRI, potentially reducing diagnostic delays and improving treatment selection accuracy [146]. Natural language processing algorithms are increasingly being utilized to extract meaningful patterns from clinical notes and patient-reported outcomes, creating more comprehensive patient profiles for treatment planning [147].

Large language models have demonstrated superior performance in sentiment analysis for pain expression [91]. Furthermore, predictive analytics models are being developed to forecast treatment outcomes and identify patients at risk of chronic pain development, enabling proactive intervention strategies [148].

Such innovative technological frameworks demonstrate significant potential in minimizing unnecessary diagnostic procedures, reducing healthcare costs, and moving away from conventional trial-and-error therapeutic approaches toward precision medicine. Systematic reviews have highlighted the growing evidence base supporting AI applications in back pain outcomes and clinical classification approaches [55]. An expanding corpus of research evidence continues to validate the efficacy of AI-driven solutions in enhancing clinical decision-making processes, improving patient outcomes, and optimizing resource allocation in LBP management.

However, the implementation of AI-driven solutions in LBP management faces several critical challenges that must be carefully addressed. Data bias represents a fundamental concern, as algorithms may perpetuate healthcare disparities when trained on datasets that underrepresent certain demographic groups, potentially leading to suboptimal treatment recommendations for vulnerable populations [149]. Cybersecurity vulnerabilities pose significant risks to patient privacy, as the valuable health data processed by AI systems makes them attractive targets for cyberattacks [150]. Finally, ethical considerations surrounding algorithmic transparency remain paramount, as clinicians and patients must understand how AI systems arrive at recommendations to maintain trust and enable informed decision-making [151]. The integration between AI and ML for the assessment and management of LBP using multimodal data to support personalized and predictive care is reported in Figure 4.

AI implementation in LBP management faces critical challenges threatening clinical translation. Algorithm bias from training on non-representative datasets may perpetuate healthcare disparities, particularly affecting minority populations underrepresented in research. The “black box” nature of deep learning models limits clinical interpretability and trust, as clinicians cannot understand decision rationales [152]. Data quality issues, including missing values, measurement errors, and inconsistent documentation, compromise model reliability. The need for large, annotated datasets for training is difficult to fill given privacy regulations and data fragmentation across healthcare systems. Model performance often degrades when applied to new clinical settings due to distribution shift. Regulatory approval processes for AI-based medical devices remain unclear and lengthy. Additionally, the medical-legal implications of AI-assisted decision-making, including liability attribution in case of errors, remain unresolved.

## 4. Discussion

This comprehensive review demonstrates that bioengineering technologies are fundamentally transforming the landscape of LBP assessment and rehabilitation, offering unprecedented opportunities for objective measurement, personalized treatment, and mechanistic understanding of this complex condition. The convergence of multiple technological domains, from wearable sensors and advanced imaging to AI-driven analytics and adaptive robotics, represents a paradigm shift from traditional, subjective clinical approaches toward data-driven, precision medicine frameworks. Traditional assessment methods, relying primarily on patient-reported outcomes and static imaging, have consistently failed to capture the dynamic, multifactorial nature of LBP [6]. Our analysis reveals that technologies such as IMUs and sEMG provide continuous, quantitative data that correlate more strongly with functional outcomes than conventional measures. This objectification of assessment is particularly crucial given that structural abnormalities on imaging frequently show poor correlation with clinical symptoms, leading to potential misdiagnosis and inappropriate treatment selection. Recent evidence suggests that movement variability and compensatory patterns, detectable only through high-resolution kinematic analysis, may be more predictive of chronicity than traditional clinical tests [57,61]. This finding challenges the conventional focus on range of motion and strength deficits, suggesting that motor control quality may be the primary therapeutic target. However, the clinical translation of these insights remains limited by the technical challenges detailed throughout this review, particularly sensor drift, placement variability, and the need for specialized expertise in data interpretation.

The heterogeneity of LBP, with its varied etiologies and individual responses to treatment, has long frustrated standardized therapeutic approaches. Bioengineering technologies offer the potential for truly personalized interventions through several mechanisms. First, multimodal data integration enables phenotyping of patients based on biomechanical, neuromuscular, and imaging signatures rather than purely symptomatic presentation [55]. Second, adaptive control systems in robotic devices and AI-driven exercise prescription can dynamically adjust therapeutic parameters based on real-time patient response [135]. Third, digital twin technology promises patient-specific simulation of treatment outcomes, though this remains largely theoretical [143]. Nevertheless, achieving meaningful personalization faces substantial barriers. The “curse of dimensionality” in multimodal data analysis risks creating overly complex models that fail to generalize across populations [55]. Furthermore, the lack of standardized protocols for data collection and analysis means that personalization algorithms developed in one setting may not transfer to another. Most critically, personalized approaches assume that individual biomechanical patterns are stable enough to guide treatment, yet emerging evidence suggests significant day-to-day variability in movement strategies among CLBP patients [25], potentially undermining the reliability of single-assessment personalization strategies.

While technological sophistication continues to advance, patient acceptance and adherence remain critical limiting factors. Our review identifies consistent challenges across multiple technologies: VR-induced cybersickness affecting up to 60% of users, discomfort from robotic exoskeletons, and frustration with technical difficulties in digital platforms. These issues are particularly pronounced in older adults, who represent the largest demographic affected by chronic LBP yet often have limited digital literacy [116]. The paradox of increasing technological complexity and potentially reducing therapeutic effectiveness deserves careful consideration. Simple interventions with high adherence may ultimately prove more effective than sophisticated systems with poor compliance. The 50% dropout rates reported in tele-rehabilitation studies suggest that technology alone cannot solve the adherence challenge [134]. Future developments must prioritize user experience design, considering not just technical capabilities but also psychological factors such as perceived ease of use, therapeutic alliance maintenance, and motivation enhancement.

The economic implications of bioengineering integration into LBP care present both opportunities and challenges. While technologies such as tele-rehabilitation promise reduced healthcare costs through decreased travel and facility requirements [131], the initial investment for advanced systems remains prohibitive. Robotic rehabilitation systems costing $50,000–$500,000 are beyond the reach of most clinical practices, potentially creating a two-tiered system where advanced treatments are available only in well-resourced centers. Moreover, the digital divide threatens to exacerbate existing healthcare disparities. Patients without reliable internet access, appropriate devices, or digital literacy—often those from lower socioeconomic backgrounds who already experience higher LBP prevalence—may be excluded from digital health innovations [132]. This raises ethical questions about the equitable distribution of technological advances and the responsibility of healthcare systems to ensure universal access to evidence-based treatments.

The rapid acceleration of technological development has outstripped regulatory frameworks, creating uncertainty for clinicians and healthcare organizations. AI-based diagnostic tools and adaptive robotic systems occupy regulatory gray areas, with unclear pathways for approval and liability concerns in case of adverse events [151]. The lack of specific reimbursement codes for many bioengineering interventions further impedes clinical adoption, as healthcare providers cannot sustain services without appropriate compensation.

The successful integration of bioengineering technologies requires careful consideration of existing clinical workflows and practitioner competencies. Physical therapists, the primary providers of LBP rehabilitation, typically receive limited training in advanced technology use and data interpretation. The time required for device setup, data analysis, and documentation may reduce direct patient contact time, potentially compromising the therapeutic relationship that remains central to successful rehabilitation [133]. Rather than replacing traditional approaches, bioengineering tools should augment clinical decision-making and enhance therapeutic precision. The most promising implementation models combine technological objectivity with clinical expertise and patient-centered care. For instance, IMU data might inform exercise prescription while maintaining hands-on manual therapy components. This hybrid approach respects the multidimensional nature of LBP, addressing both mechanistic dysfunction and psychosocial factors.

### Challenges, Future Directions, and Limitations

Despite rapid advancements, the integration of bioengineering technologies into routine care for LBP remains challenged by practical, technical, and ethical limitations. While the cost and logistical complexity of high-fidelity systems limit their widespread availability [152,153], the technology-specific challenges detailed throughout this review, ranging from sensor drift in IMUs and signal crosstalk in sEMG to cybersickness in VR systems and muscle fatigue in NMES, underscore the multifaceted nature of implementation barriers. The interpretation of multimodal data streams necessitates expertise in biomedical informatics and robust computational infrastructure, which are not universally accessible [154], a challenge compounded by the lack of standardized protocols for sensor placement, radiomic feature extraction, and robotic therapy outcomes. Moreover, the absence of standardized validation protocols and outcome measures impedes cross-comparison and limits generalizability to diverse patient populations [155], particularly problematic given the demonstrated performance degradation of AI models when applied across different clinical settings and the poor inter-scanner reproducibility of radiomic signatures. Patient adherence, especially in unsupervised or home-based digital rehabilitation programs, remains an enduring challenge, with motivation and digital literacy emerging as important moderating factors [156], as evidenced by dropout rates exceeding 50% in tele-rehabilitation and the technology gap affecting older populations in VR-based interventions. Ethical concerns surrounding data security, algorithmic transparency, and clinician accountability in AI-assisted decision-making are also pressing and require stringent oversight frameworks [157], particularly given the “black box” nature of deep learning models and unresolved medical-legal implications. These cumulative limitations highlight that successful clinical translation requires not only technological refinement but also systematic approaches to standardization, training, and equitable access to ensure that these promising innovations can effectively serve the diverse LBP population.

Looking forward, the next decade is poised for deeper convergence between bioengineering and precision medicine. Integrated platforms combining motion analysis, EMG, imaging biomarkers, and psychological profiling could enable individualized diagnostic and therapeutic trajectories [132]. Predictive models, powered by AI, may identify early indicators of pain exacerbation, adherence decline, or functional recovery potential [142,144]. Furthermore, bioengineering tools are expected to support regenerative medicine applications, offering quantitative tracking of biologic therapies such as mesenchymal stem cell injections or platelet-rich plasma interventions through imaging and molecular biomarkers [158]. The integration of bioengineered outputs into electronic health records will allow real-time decision support and personalized rehabilitation planning. Smart rehabilitation environments, equipped with ambient sensors and adaptive feedback systems, may autonomously guide patient activities and monitor outcomes, redefining standards in physical therapy [158]. Realizing this vision demands sustained interdisciplinary collaboration among engineers, clinicians, and data scientists to ensure that these innovations are rigorously validated and translated into scalable, equitable clinical practice.

Limitations: This narrative review, while developed in accordance with SANRA guidelines to ensure methodological rigor and transparency, is inherently limited by its non-systematic approach, which may introduce selection bias and affect reproducibility. The absence of a formal search strategy and critical appraisal protocol restricts the capacity to comprehensively capture the full spectrum of relevant evidence. Moreover, the included studies exhibit significant heterogeneity in terms of design, bioengineering technologies applied, and outcome measures employed, thereby limiting comparability and generalizability of the findings. Finally, given the fast-paced innovation within the field of bioengineering, certain conclusions may be subject to rapid obsolescence as new data emerge.

## 5. Conclusions

Bioengineering plays a pivotal role in advancing the assessment and rehabilitation of LBP by offering objective, personalized, and data-driven tools. From wearable sensors and robotic supports to AI-powered decision systems, these technologies enhance diagnostic precision and therapeutic outcomes. While challenges remain in terms of cost, integration, and standardization, the potential for transformative impact is substantial. Future research should prioritize accessibility, ethical deployment, and the development of interoperable systems that bridge the gap between cutting-edge innovation and clinical routine.

## Figures and Tables

**Figure 1 bioengineering-12-00900-f001:**
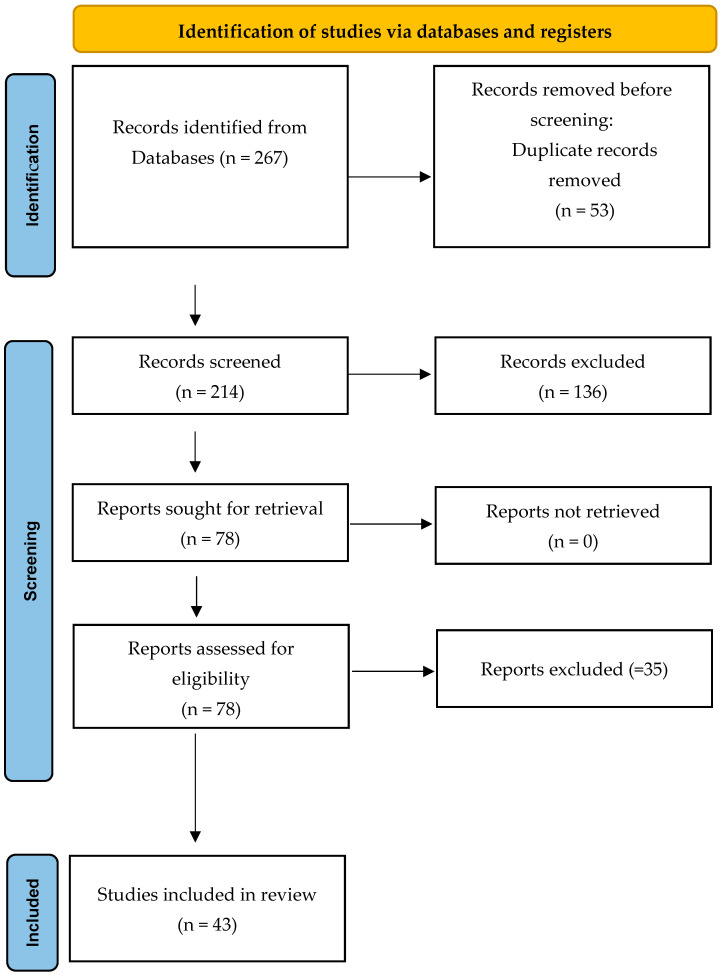
PRISMA flow diagram illustrating the selection process of the literature related to the topic.

**Figure 2 bioengineering-12-00900-f002:**
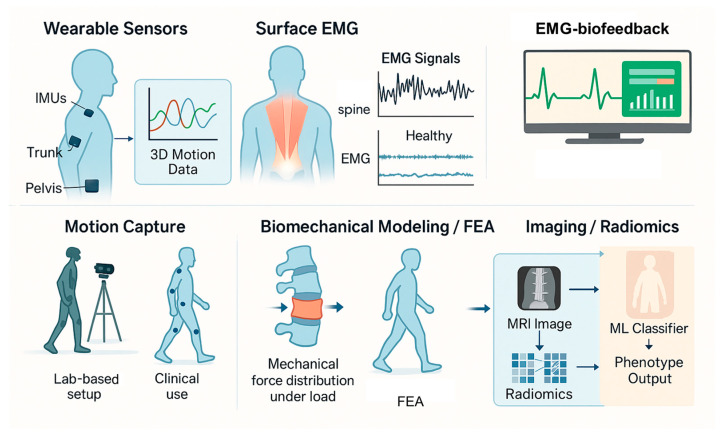
Bioengineering tools used in the assessment of LBP. The figure highlights key technologies including wearable inertial sensors (IMUs) for motion tracking, surface electromyography (sEMG) for evaluating muscle activation and fatigue, motion capture systems for kinematic analysis, biomechanical modeling and finite element analysis (FEA) for simulating spinal load distribution, and advanced imaging with radiomics and machine learning classifiers for phenotype identification and personalized assessment.

**Figure 3 bioengineering-12-00900-f003:**
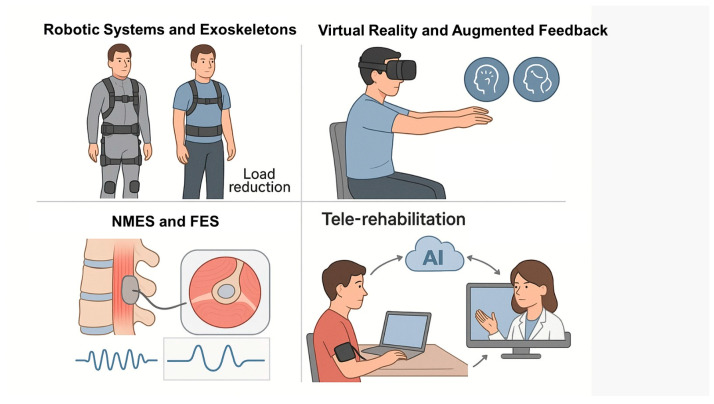
Bioengineering approaches in the rehabilitation of LBP, including robotic systems and exoskeletons designed to reduce spinal load; virtual reality and augmented feedback to enhance motor engagement and reduce kinesiophobia; neuromuscular electrical stimulation (NMES) and functional electrical stimulation (FES) targeting paraspinal muscles; and AI-powered tele-rehabilitation platforms enabling remote, personalized care.

**Figure 4 bioengineering-12-00900-f004:**
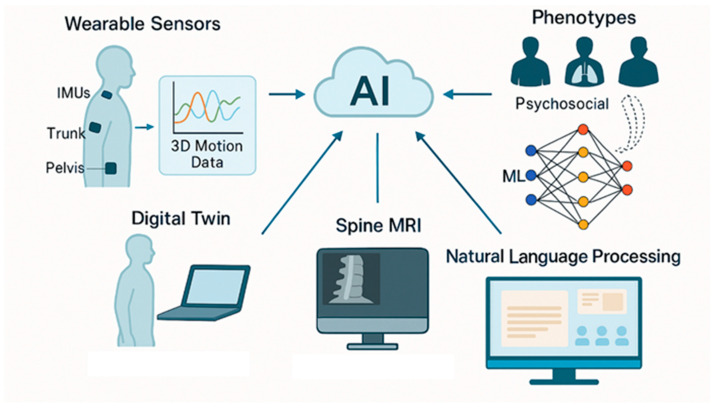
Integration of AI and ML in LBP management. Multimodal data sources—such as wearable sensor outputs (e.g., inertial measurement units), spine MRI, digital twin simulations, and natural language processing of clinical notes—are processed by AI systems to classify patient phenotypes, predict outcomes, and optimize personalized rehabilitation strategies.

**Table 1 bioengineering-12-00900-t001:** Primary division of the most relevant bioengineering domains emerging from the literature related to the use of advanced technologies in LBP patients.

Bioengineering Domain	Applications in LBP	N. of Relevant Studies
Wearable Sensors	Real-time monitoring of posture and movement	8
Surface EMG	Assessment of muscle activity and fatigue	6
Motion Capture	Quantification of functional tasks and gait	5
Biomechanical Modeling	Simulation of spinal loads and tissue stress	6
Robotic Rehabilitation Systems	Automated, adaptive therapeutic support	5
Advanced Imaging	Quantitative biomarkers for diagnosis and progression	7
AI/Machine Learning	Predictive analytics, phenotyping, decision support	6

## Data Availability

Data are available upon reasonable request to the corresponding author.

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
