# Peer review of "Bioengineering Support in the Assessment and Rehabilitation of Low Back Pain"

_bioengineering, 2025, doi:10.3390/bioengineering12090900_

Round 1
Reviewer 1 Report
Comments and Suggestions for Authors
The paper offers a thorough overview of current research on low back pain, covering the technologies used for its assessment and treatment. It also outlines directions for future research in the field. This review is particularly valuable for new researchers entering the area. No concerns were found in the review.
Author Response
The paper offers a thorough overview of current research on low back pain, covering the technologies used for its assessment and treatment. It also outlines directions for future research in the field. This review is particularly valuable for new researchers entering the area. No concerns were found in the review.
We sincerely thank the reviewer for the positive feedback and for recognizing the value of our work in providing an overview of current technologies for the assessment and treatment of low back pain.
Reviewer 2 Report
Comments and Suggestions for Authors
This narrative review examines current bioengineering applications in both diagnostic and therapeutic domains. For assessment, technologies such as wearable inertial sensors, three-dimensional motion capture systems, surface electromyography, and biomechanical modeling provide real time, quantitative insights into posture, movement patterns, and muscle activity. The innovation of this article is a very interesting topic. However, there are still some problems that need to be modified, as follows:
- In "Introduction" 、“Materials and Methods” and “Results” section Related Works, I feel the current coverage of the state of the art is not satisfactory as the related work section does not cover many contributions that likely provide the building blocks of the proposed approach. For example, a) Adaptive human-robot interaction torque estimation with high accuracy and strong tracking ability for a lower limb rehabilitation robot, IEEE/ASME Transactions on Mechatronics, 2024. b) ViT-based Terrain Recognition System for wearable soft exosuit[J]. Biomimetic Intelligence and Robotics, 2023. c) Human–robot interface based on sEMG envelope signal for the collaborative wearable robot[J]. Biomimetic Intelligence and Robotics, 2023.
- There are some mistakes including grammar, words and English expression in this paper. Please check the overall paper carefully.
3. Discussion section should be added and discussed in detail. The Discussion section should be edited in a more highlighting, argumentative way.
Author Response
This narrative review examines current bioengineering applications in both diagnostic and therapeutic domains. For assessment, technologies such as wearable inertial sensors, three-dimensional motion capture systems, surface electromyography, and biomechanical modeling provide real time, quantitative insights into posture, movement patterns, and muscle activity. The innovation of this article is a very interesting topic. However, there are still some problems that need to be modified, as follows:
- In "Introduction" 、“Materials and Methods” and “Results” section Related Works, I feel the current coverage of the state of the art is not satisfactory as the related work section does not cover many contributions that likely provide the building blocks of the proposed approach. For example, a) Adaptive human-robot interaction torque estimation with high accuracy and strong tracking ability for a lower limb rehabilitation robot, IEEE/ASME Transactions on Mechatronics, 2024. b) ViT-based Terrain Recognition System for wearable soft exosuit[J]. Biomimetic Intelligence and Robotics, 2023. c) Human–robot interface based on sEMG envelope signal for the collaborative wearable robot[J]. Biomimetic Intelligence and Robotics, 2023.
We sincerely thank the reviewer for highlighting this important point and for suggesting highly relevant topics. In response, we have substantially expanded our coverage of advanced human–robot interaction and adaptive control strategies by adding a new subsection (Advanced Human-Robot Interfaces and Adaptive Control Systems). This section now integrates and discusses the cited works on adaptive torque estimation, ViT-based terrain recognition, and sEMG envelope–based human–robot interfaces
2. There are some mistakes including grammar, words and English expression in this paper. Please check the overall paper carefully.
We thank the reviewer for this valuable observation. The manuscript has been carefully revised, and grammar, wording, and English expressions have been corrected.
- Discussion section should be added and discussed in detail. The Discussion section should be edited in a more highlighting, argumentative way.
We thank the reviewer for this valuable suggestion. In response, we have substantially expanded and restructured the Discussion section, providing a more critical and argumentative analysis of the findings.
Reviewer 3 Report
Comments and Suggestions for Authors
This manuscript is a narrative review on how recent advancement of bioengineering technologies can help to understand and manage low back pain, which is one of the most disability causing medical problems in modern society.
It is well-written manuscript covering recent advancement of bioengineering technologies comprehensively, including IMU sensors, sEMG, motion analysis, modeling and FEA, Imaging biomarkers as tools for assessment, and robotics, VR/AR, NMES and FES, tele-rehab for rehabilitation purposes.
Major issue:
While the authors summarized the utility of each bioengineering technology for assessing or rehabilitating LBP in detail, going over individual articles when necessary, it reads unusual that there are no mentions on practical limitations of each technology, which must be overcome or at least, circumvented to be applied in clinical fields. As a clinical researcher who have tried most of the technologies listed above, there were distinct limitations that hindered clinical application. Although the authors provided one paragraph, from lines 572~585, to address limitations of bioengineering technologies as a whole, how about adding specific limitations of each technology at the end of each section? I am convinced that it will draw increased attention of readers and gain more citations by other researchers, as well.
Minor issues:
I do think it is good to include NMES-FES, tele-rehab, and VR/AR in this review but how did the authors found the articles related with them even though they did not include them in the search term in lines 115~118?
In Figure 2, in the lower left corner subset, what is “Lab-basel Setup”? A typo?
Author Response
This manuscript is a narrative review on how recent advancement of bioengineering technologies can help to understand and manage low back pain, which is one of the most disability causing medical problems in modern society.
It is well-written manuscript covering recent advancement of bioengineering technologies comprehensively, including IMU sensors, sEMG, motion analysis, modeling and FEA, Imaging biomarkers as tools for assessment, and robotics, VR/AR, NMES and FES, tele-rehab for rehabilitation purposes.
Major issue:
While the authors summarized the utility of each bioengineering technology for assessing or rehabilitating LBP in detail, going over individual articles when necessary, it reads unusual that there are no mentions on practical limitations of each technology, which must be overcome or at least, circumvented to be applied in clinical fields. As a clinical researcher who have tried most of the technologies listed above, there were distinct limitations that hindered clinical application. Although the authors provided one paragraph, from lines 572~585, to address limitations of bioengineering technologies as a whole, how about adding specific limitations of each technology at the end of each section? I am convinced that it will draw increased attention of readers and gain more citations by other researchers, as well.
We sincerely thank the reviewer for this insightful comment. We fully agree that discussing the specific limitations of each bioengineering technology would strengthen the clinical relevance of our work. In response, we have added dedicated sentences at the end of each section to highlight the practical limitations and barriers to clinical translation for each modality. We believe this addition improves the balance of our review, enhances readability for clinicians and researchers, and aligns well with the reviewer’s suggestion to increase the manuscript’s applicability and citation potential.
Minor issues:
I do think it is good to include NMES-FES, tele-rehab, and VR/AR in this review but how did the authors found the articles related with them even though they did not include them in the search term in lines 115~118?
We thank the reviewer for this important observation. While these technologies were not explicitly included in our initial search terms, we identified them through citation tracking of included articles and relevant systematic reviews. This manual screening highlighted VR, AR, NMES, FES, and tele-rehabilitation as emerging modalities increasingly integrated with sensor-based monitoring and AI-driven personalization in LBP rehabilitation. To ensure comprehensive coverage, we subsequently performed a supplementary targeted search using these specific terms in combination with “low back pain.” We have clarified this two-step process in the Methods section.
In Figure 2, in the lower left corner subset, what is “Lab-basel Setup”? A typo?
We thank the reviewer for noticing this. It was indeed a typographical error. We have corrected “Lab-basel Setup” to “Lab-based Setup” in Figure 2.
Round 2
Reviewer 2 Report
Comments and Suggestions for Authors
The authors have answered my questions well. It can be accepted before correct the authors of reference 136. Thank you.